# Synthesis, Antimalarial, Antileishmanial, and Cytotoxicity Activities and Preliminary In Silico ADMET Studies of 2-(7-Chloroquinolin-4-ylamino)ethyl Benzoate Derivatives

**DOI:** 10.3390/ph16121709

**Published:** 2023-12-09

**Authors:** Joyce E. Gutiérrez, Hegira Ramírez, Esteban Fernandez-Moreira, María E. Acosta, Michael R. Mijares, Juan Bautista De Sanctis, Soňa Gurská, Petr Džubák, Marián Hajdúch, Liesangerli Labrador-Fagúndez, Bruno G. Stella, Luis José Díaz-Pérez, Gustavo Benaim, Jaime E. Charris

**Affiliations:** 1Organic Synthesis Laboratory, Faculty of Pharmacy, Central University of Venezuela, Los Chaguaramos 1041-A, Caracas 1040, Venezuela; gutierrez.joyce@gmail.com; 2Facultad de Ciencias de la Salud y Desarrollo Humano, Univesidad Ecotec, Km. 13.5 Samborondón, Guayas, Guayaquil 092302, Ecuador; 3Escuela de Medicina, Universidad Espíritu Santo, Samborondón, Guayas, Guayaquil 092301, Ecuador; estebanfernandez@uees.edu.ec; 4Unidad de Bioquímica, Facultad de Farmacia, Central University of Venezuela, Los Chaguaramos 1041-A, Caracas 1040, Venezuela; mariuacosta0103@gmail.com; 5Biotechnology Unit, Faculty of Pharmacy, Central University of Venezuela, Los Chaguaramos 1041-A, Caracas 1040, Venezuela; michaelfarmacia@gmail.com; 6Institute of Molecular and Translational Medicine, Faculty of Medicine and Dentistry, Palacky University, Hněvotínská 1333/5, 779 00 Olomouc, Czech Republic; juanbautista.desanctis@upol.cz (J.B.D.S.); sona.gurska@upol.cz (S.G.); petr.dzubak@upol.cz (P.D.); marian.hajduch@upol.cz (M.H.); 7Unidad de Bioquímica de Parásitos y Señalización Celular, Instituto de Estudios Avanzados (IDEA), Caracas 1080, Venezuela; g12liesa@gmail.com (L.L.-F.); brunogiancarlostellachacin@gmail.com (B.G.S.); ljdp0754@gmail.com (L.J.D.-P.); gbenaim@gmail.com (G.B.); 8Instituto de Biología Experimental, Facultad de Ciencias, Central University of Venezuela, Caracas 1040, Venezuela

**Keywords:** malaria, leishmaniasis, cytotoxicity, ADMET, chloroquine, aminoalkylbenzoates

## Abstract

A series of heterocyclic chloroquine hybrids, containing a chain of two carbon atoms at position four of the quinolinic chain and acting as a link between quinoline and several benzoyl groups, is synthesized and screened in vitro as an inhibitor of β-hematin formation and in vivo for its antimalarial activity against chloroquine-sensitive strains of *Plasmodium berghei* ANKA in this study. The compounds significantly reduced haeme crystallization, with IC_50_ values < 10 µM. The values were comparable to chloroquine’s, with an IC_50_ of 1.50 ± 0.01 µM. The compounds **4c** and **4e** prolonged the average survival time of the infected mice to 16.7 ± 2.16 and 14.4 ± 1.20 days, respectively. We also studied the effect of the compounds **4b**, **4c,** and **4e** on another important human parasite, *Leishmania mexicana*, which is responsible for cutaneous leishmaniasis, demonstrating a potential leishmanicidal effect against promasigotes, with an IC_50_ < 10 µM. Concerning the possible mechanism of action of these compounds on *Lesihmania mexicana*, we performed experiments demonstrating that these three compounds could induce the collapse of the parasite mitochondrial electrochemical membrane potential (Δφ). The in vitro cytotoxicity assays against mammalian cancerous and noncancerous human cell lines showed that the studied compounds exhibit low cytotoxic effects. The ADME/Tox analysis predicted moderate lipophilicity values, low unbound fraction values, and a poor distribution for these compounds. Therefore, moderate bioavailability was expected. We calculated other molecular descriptors, such as the topological polar surface area, according to Veber’s rules, and except for **2** and **4i**, the rest of the compounds violated this descriptor, demonstrating the low antimalarial activity of our compounds in vivo.

## 1. Introduction

Humans become infected with malaria when bitten by female Anopheles mosquitoes that carry *Plasmodium* parasites. Six parasite species cause malaria in humans, with *P. falciparum* and *P. vivax* being the two most dangerous, which are often fatal, and more than 40% of the world’s population are now at risk of malaria infection [1]. By 2020, there had been an estimated 241 million new malaria cases worldwide and 627,000 malaria deaths [2].

Climate change and environmental issues have been linked to the resurgence of malaria in some parts of the world; climate change and environmental issues have led to changes in the life cycle, duration of activity, and proliferation of Anopheles [3]. Another factor to consider is the increasing resistance of Anopheles strains to residual insecticides. However, recent scientific advances have been providing new tools for malaria control, among which we can mention the first effective malaria vaccine, RTS, S/AS01 (RTS, S), which was approved by the World Health Organization in October 2021 [4]. In addition, a clinical trial tested a new long-term chlorfenapyr insecticidal net that could mitigate the effects of insecticide resistance in mosquitoes [5]. Another factor contributing to the resurgence of malaria is its resistance to drugs used in the clinic for its treatment and prevention, such as chloroquine (CQ) and artemisinin and its derivatives, among others, as seen in Figure 1 [6,7].

Another parasitic disease that we will address in this study is leishmaniasis, a protozoan parasite group that includes more than 20 *Leishmania* species. After being inoculated in the bloodstream by the sandfly, these parasites are actively absorbed by circulating macrophages and then transform into amastigotes. After reaching a critical number, the host cell is disrupted, discharging the parasites and invading other macrophages, thus repeating the cycle and resulting in infection [8]. The three main forms of leishmaniasis are cutaneous (CL), mucocutaneous (MCL), and visceral (VL), also known as kala-azar. Approximately 0.9–1.6 million new cases are reported every year, and the disease’s mortality rate is between 20,000 to 30,000 cases per year. CL is the most common form of leishmaniasis, leading to 600.000 to 1 million new infections annually [9]. The clinical symptoms of CL are restricted to the skin lesions or mucosal affectations for MCL. However, VL is characterized by severe organic symptoms and can lead to death [10,11].

A limited number of drugs are available for treatment against leishmaniasis, and the recommended first-line therapies for leishmaniasis include pentavalent antimony compounds, such as sodium stibogluconate and meglumine antimoniate; these drugs have secondary effects on the renal, cardiac, and hepatic systems. Leishmaniasis’ resistance to antimonial pentavalent drugs is also an important factor that limits this treatment. Another disadvantage of antimonial therapy is the requirement for relatively long-term parenteral administration to obtain optimal results. The second-line treatments include pentamidine and amphotericin B, as seen in Figure 2. The polyene antibiotic amphotericin B constitutes an alternative treatment for visceral leishmaniasis, mainly in its liposomal presentation (AmBisome), with reduced toxicity and high effectiveness. It is also costly and sometimes still toxic, resulting in renal failure. Miltefosine, an orally active alkyl-lysophospholipid with potent anti-*Leishmania* activity, represents a major advance in the treatment of leishmaniasis since it is the only oral drug currently approved. Nevertheless, miltefosine presents a series of adverse effects, such as teratogenicity, which limits its use during pregnancy. This drug also apparently induces the potential development of resistance [12,13,14,15,16].

Chloroquine and its derivatives, whose central structures are constituted by a quinoline, have always attracted chemists and biologists because of their diversity of chemical and pharmacological properties, for example as antimalarial, anticancer, antibacterial, antiviral, antifungal, antituberculous, antioxidant, antiasthmatic, antileishmanial, antipsychotic, antiglaucomatous, and cardiovascular agents (Figure 3) [17,18,19]. Great efforts have been made to find new natural or synthetic structures to help solve the health problems mentioned above.

For antimalarials containing 4-aminoquinolines, structure–activity relationships (SAR) indicate that the 7-chloro-4-aminoquinoline core is critical. Quinoline nuclei have been shown to inhibit hematin formation and allow drug accumulation at the target site through hydrogen bonding, π-π-stacking, and acid–base interactions [20,21,22,23,24]. An alternative strategy for developing new antimalarial drugs is the hybridization approach, which has proven successful in developing effective antiplasmodial agents [25,26,27,28,29,30,31,32,33]. The term hybrid drug generally refers to the combination of two or more biologically active molecules or two pharmacophore groups of different compounds in one molecule. The combination results in a new molecule, now referred to as a hybrid, which may be more potent, less potent, or identical to its precursor compound [34,35]. In hybrid molecules, a linker may be present (hybrids with linkers) or not (fused hybrids), or they may be linked together (chimeric) [36,37,38,39].

Our research group also explored the hybridization method in previous studies to obtain many different hybrids against *P. berghei* and cancer cell lines by some specific modifications. Compared with CQ, some compounds showed an excellent in vitro activity and significant suppression of parasitaemia and cell viability [40,41,42,43,44].

Based on the results obtained by our group, supported by the reports of SAR, it was concluded that the nitrogen in the quinoline ring and the side chain plays a vital role in the antiparasitic and anticancer properties [45]. Given these results and other previously reported observations, we present the synthesis of thirteen compounds modified at position four of the quinoline ring CQ with a chain of two carbon atoms acting as a link between quinoline and the variety of benzoyl groups predicted for antiparasitic activity and anticancer activity. Different substituents are used on the aromatic benzoyl ring to synthesise various compounds. In addition to inhibiting haeme polymerisation in vitro, we report the antiproliferative activity of the synthesized compounds in mice infected with *P. berghei* in vivo. We also studied the effect of these compounds on another important human parasite, *Leishmania mexicana*, the causative agent of cutaneous leishmaniasis (CL) and mucocutaneous leishmaniasis (MCL). Experiments demonstrated that three compounds could induce the collapse of the parasite mitochondrial electrochemical membrane potential (Δφ) and the poor antiproliferative effect on eight human cancer cell lines and two non-malignant cell lines in vitro.

In silico profiling of absorption, distribution, metabolism, excretion, and toxicity (ADME/Tox) was also performed to predict the molecular properties of novel compounds’ pharmacokinetic aspects.

## 2. Results and Discussion

### 2.1. Chemistry

The compound 2-(7-Chloroquinolin-4-ylamino)ethanol **2** was prepared from an excess of 2-amino ethanol, triethylamine, and 4,7-dichloroquinoline **1** in a good yield of 95%. As a result of coupling reactions between **2** and substituted benzoic acids in combination with EDCI and DMAP [46], the final compounds **4a**–**m** could be synthesized. Following purification by recrystallization or column chromatography, the compounds were isolated in good to excellent yields (60–94%) (Figure 1). IR, ^1^H NMR, ^13^C NMR, DEPT 135°, COSY, and HETCOR spectroscopy were used to characterise the compounds (Appendix A). A proposal for the reaction mechanism in the formation process of the final compounds **4a**–**m** is described in Figure 2.

For **4a**–**m**, the IR spectra show peaks in the range of 3239–2960 cm^−1^ for the NH, C–H stretching vibrations, and 1733–1692 cm^−1^ for the C=O stretching vibrations. Bands around 1584 and 1520 cm^−1^ were produced by the N–H stretching vibrations originating from amino groups.

According to their chemical shifts, multiplicities, and coupling constants, the signal of the protons of each compound was checked in the ^1^H NMR spectra. Hs9 shows aliphatic signals from 3.64 to 3.84 ppm, while Hs10 shows aliphatic signals from 4.40 to 4.74 ppm, and the NH signals appear as a singlet at 4.26–5.98 ppm. As a result of the quinoline moiety, proton H3 appeared to be a doublet (d, *J* = 5 Hz). In contrast, proton H6 appeared to be a double doublet about 7.3 ppm (dd, *J* = 8 and 2 Hz), proton H5 was assigned a doublet around 7.7 ppm (d, *J* = 8 Hz), proton H8 was assigned a doublet around 7.9 ppm (d, *J* = 2 Hz), and proton H2 was assigned a doublet around 8.5 ppm (d, *J* = 5 Hz). In the ^1^H NMR spectra, a variety of signal patterns were detected in the aromatic region ranging from *δ*H 6.5 to 8.1 ppm, which were indicative of the substitution pattern of each aromatic ring. In addition, the chemical shifts of the carbon atoms were observed and confirmed by the ^13^C NMR spectra analysis. Peaks around 43, 62, 99, 121, 125, 128, 152, and 166 ppm, for instance, can be attributed to C9, C10, C3, C5, C6, C8, and C=O, respectively. The experimental section summarised the analytical and spectroscopic data of each compound.

### 2.2. In Silico Analysis

Using *SwissADME*, drug-likeness descriptors were calculated using Lipinski and Veber rules [47,48]. The use of these descriptors and results indicating non-violation of Lipinski’s rule of five could predict good absorption or permeation of each molecule; however, there are some exceptions to the “rule of five” for certain classes of drugs (i.e., vitamins, antifungals, antibiotics, and cardiac glycosides). Table 1 summarises these results. The compounds **2**, **4a**–**m**, in general, did not violate these rules according to these criteria, except CQ [diphosphate] CQ(DPh) [48].

According to the prediction, the bioavailability score of all compounds was about 0.55, except for CQ(DPh), which had a value of 1.00. In the BOILED-Egg illustration (Figure 4A), all combinations are predicted to demonstrate high gastrointestinal (GI) absorption; however, for CQ(DPh) this was low or out of range. In particular, the compounds **4f**, **k**, and **m** are expected to be passively absorbed by the gastrointestinal tract. In addition, the compounds **2**, **4a**–**e**, **g**–**j**, and **l** are predicted to permeate the blood–brain barrier passively. All compounds demonstrated a TPSA value < 140 Å^2^, thus indicating their good oral bioavailability, except CQ(DPh) with a TPSA value of 203.30 Å². Furthermore, all compounds displayed moderate drug-likeness scores ranging from the compounds **2** = −0.13, **4c** = 0.66, and **4e** = 0.49, probably due to the absence of a more basic moiety on position four of the quinoline core, except CQ(DPh) with a score of 1.00 (Figure 4B–E).

Figure 5 summarises the main ADME parameters in the radar representations for the compounds **2**, **4c**, **4e**, and CQ(DPh) properties, including lipophilicity, size, polarity, solubility, saturation, and flexibility. The colour zone is the suitable physicochemical space for oral bioavailability, and the red bold line represents values of the calculated properties of the analysed molecule. CQ(DPh) exhibited all of the properties characteristic of orally available drugs.

However, CQ(DPh) had a low predicted absorption in the gastrointestinal tract. Motivated that the water solubility of a drug has been related to the pharmaceutical form, route of administration, and absorption, we proceeded to evaluate the water solubility of the compounds **2**, **4c**, **4e**, and CQ(DPh) using the *SwissADME* program. The program uses two topological methods to predict aqueous solubility: the ESOL (estimated solubility) model [49], the model incorporating the effect of topographical polar surface area [50], and a third predictor for solubility developed by SILICOS-IT. All predicted values are shown as decimal logarithms of the molar solubility in water (log S). The values obtained for the compounds **2**, **4c**, **4e**, and CQ(DPh) are shown in Table 2. CQ(DPh) has a higher water solubility with a log S (ESOL) of −1.59 than the compounds **2**, **4c**, and **4e** (log S (ESOL) of −3.24, −4.85, and −4.85, respectively). An antiparasitic compound needs to reach the target to produce a positive effect. Thus, the positive outcome observed when administering chloroquine bisphosphate by the intraperitoneal route could be attributed to the polarity and higher solubility in water. The compound **2** showed a good water solubility, yet it was not evaluated in vivo because it had an IC_50_ of 5.06 ± 0.31 µM as an inhibitor of the formation of β-hematin. In contrast, **4c** and **4e** showed moderate solubility in water, which could be related to the high degree of unsaturation, size, and flexibility in these two molecules, not favouring intraperitoneal administration, the desired effect in vivo. These two compounds may be absorbed orally in the small intestine based on the favourable solubility values.

Using *pkCSM-pharmacokinetics*, several molecular descriptors that complement our in silico evaluation were calculated, including the percentage of the compounds **2**, **4a**–**m** absorbed through the human intestine (%HIA). Appropriate values were considered in the range of 88.32% to 95.38%. On the other hand, for CQ(DPh), this value was 31.34, making it a poor oral administration candidate. In this study, the descriptors fraction unbound compound (FU) and total clearance (CLtot) were used to predict the distribution [51,52,53]. FU affects renal glomerular filtration and hepatic metabolism, affecting drug volume distribution, efficiency, and clearance. In this study, the compounds **2** and **4a**–**m** FU had values between 0 and 0.318, and CQ(DPh) was 0.424. The total clearance (CLtot) descriptor was used to estimate the amount of excretion, which ranged from 0.016 to 0.379 mL/min/kg, where CQ(DPh) was 0.314, except for the compounds **4f** and **g,** which had values of 0.638 and 0.553 mL/min/kg, respectively.

The hepatotoxicity and oral toxicity LD_50_ values of the compounds **2**, **4a**–**m**, and CQ(DPh) were predicted [51,54]. The possibility that the compounds **2**, **4a**–**m**, and CQ(DPh) are substrates or inhibitors of selected human transporters was analysed. The targets can be P-glycoproteins vital in cell distribution and excretion. The ADMET predictor software https://biosig.lab.uq.edu.au/pkcsm/license (accessed on 15 March 2023) *pkCSM-pharmacokinetics* revealed that all of the compounds except **2**, **4h**, **4k**, and CQ(DPh) are not P-glycoprotein substrates. Concerning inhibitory action, most of the compounds examined are inhibitors of P-glycoprotein, except the compounds **2** and CQ (DPh). This effect is related to the bile salt export pump (BSEP), expressed almost exclusively in the liver, whose inhibition is associated with hepatotoxicity. BSEP inhibition can result in the accumulation of bile salts in the liver, leading to the induction of liver injury. In our case, most compounds exhibit possibilities to inhibit BSEP and thus induce potential hepatotoxicity, except the compounds **2** and CQ(DPh).

As determined by inhibition of the main cytochromes (CYPs) of the P450 superfamily (CYP2D6, CYP3A4, CYP1A2, CYP2C19, and CYP2C9) (Table 3), metabolism was estimated by the web tool *pkCSM-pharmacokinetics* [55,56,57,58]. Except for **2, 4f, g,** it has been predicted that the compounds **4a**–**e**, **h**–**m** may inhibit CYP3A4, CYP1A2, CYP2C19, and CYP2C9. On the other hand, the results showed that the compounds do not inhibit the CYP2D6 isoform. The predicted cytochrome P450 metabolism profile is more favourable for the compound **2**, since it is not expected to inhibit the evaluated enzyme isoforms, which agrees with the values shown by chloroquine. The inhibition of the CYPs isoforms may lead to the inefficient or toxic effects of drugs.

### 2.3. Antimalarial Activity

As part of our efforts to identify potential antimalarial agents among the 4-aminoquinoline derivatives **2**, **4a**–**m**, the fourteen derivatives were evaluated in vitro as inhibitors of β-H formation and in vivo in a murine model, and their results are presented in Table 4. Compounds significantly reduced haeme crystallization to a half maximal inhibitory concentration of less than 10 µM (IC_50_ < 10 µM). The compounds **4c** and **4e** inhibited haeme crystallization with IC_50s_ of 2.10 ± 0.48 µM and 1.81 ± 0.83 µM, respectively, compared to CQ (0.18 ± 0.03 µM).

The compounds **4c** and **4e** were evaluated in mice infected with *P. berghei* ANKA, a strain of murine malaria susceptible to CQ(DPh). The compounds **4c** and **4e** were assessed on the fourth day post-infection by evaluating their ability to increase mouse survival and reduce parasitaemia in vivo compared to untreated controls. The compounds **4c**, **4e** (25 mg/kg), or CQ (25 mg/kg) were administered intraperitoneally (ip) once daily in mice following previously reported protocols [42,59,60,61]. The Institute of Immunology Bioethical Committee approved the study (AS 161019). We followed the guidelines for Laboratory Animal Research of the National Research Council (ILAR).

The structures **4c** and **4e** used as monotherapies prolonged the average survival time of the infected mice to 16.71 ± 2.16 days and 14.43 ± 1.20 days, respectively. However, they were unable to decrease or delay the development of malaria. CQ(DPh) prolonged survival to 30 days in mice and reduced malaria development by 0.28 ± 0.15%.

In addition, both compounds were evaluated for their haemolytic response [62]. In mouse red globules at a concentration of 1 mM, haemolysis was less than 5%, showing these compounds do not have a significant lytic effect on mice’s RBCs. All compounds were highly effective as β-H inhibitors; the compounds (**4c**, **d**), doubly substituted with OMe groups showed lower IC_50_ values than compounds substituted with mono-OMe, tri-OMe, mono-halogen, or mono-alkyl groups. We inferred from the marginal activity observed in vivo that these compounds may have a moderate solubility in water, a moderate partition coefficient, a low unbound fraction, a strong inhibition of the main cytochromes (CYPs) of the P450 superfamily, and hepatotoxicity at the dose administered to each group of mice based on the predictive values obtained in silico.

### 2.4. Antileishmanial Activity

Based on the results obtained on malaria, we decided to determine the possible effect of the compounds **4b**, **4c**, and **4e** on the growth of promastigotes of *Leishmania mexicana* parasites. The IC_50_ values obtained with these three compounds, depicted in Table 5, were relatively low since the compounds **4b**, **4c**, and **4e** showed an IC_50_ of 8.09 ± 1.47, 8.46 ± 1.86, and 5.67 ± 2.15, respectively (Table 5), thus promising to be potential candidates against leishmaniasis. We used Triton X-100 at a concentration of 0.1% as a positive control since this detergent dissolves the plasma membrane and kills the parasite. We also used in these experiments the compound SQ 109, a well-known compound which induces *Leishmania mexicana* promastigotes’ death with an IC_50_ of 0.53 ± 0.06 µM [15]. We then looked for these compounds’ possible mechanism of action on *L. mexicana*. Our first approach was to determine if these compounds affect the large unique mitochondrion present in these organisms, which is well known to be involved in the bioenergetics and the intracellular Ca^2+^ regulation of these parasites [8], constituting an attractive possible therapeutic target. As can be seen in Figure 6, all three compounds highly affected the mitochondrial function. This event could be observed by releasing the previously accumulated rhodamine 123 fluorescent indicator due to mitochondrial de-energization. Thus, the compounds can induce the collapse of the mitochondrial electrochemical membrane potential (ΔΨ_m_). In these experiments, the protonophore (FCCP) was used to control mitochondrial de-energization since this drug induces the total collapse of the electrochemical H^+^ gradient in these parasites [63]. Accordingly, in these experiments, it can be observed that after the addition of the different compounds **4b**, **4c** and **4e**, a rapid increase in the fluorescence was observed, followed by an extra rise when FCCP was added after the plateau.

On the other hand, the effects of the compounds **4b** and **4e** were significantly higher when compared with the compound **4c.** This effect can also be noticed in the quantification of the fluorescence magnitude depicted in Figure 6, showing that the value obtained for **4c** is lower when compared to **4b** and **4e**. Accordingly, with these results, the effect of FCCP is higher when the compound **4c** was used due to the lower degree of de-energization induced by this compound. Curiously, there is no clear correlation between the IC_50_ values of the three compounds and their effects on mitochondrial function. Thus, the compounds **4b** and **4c** possess very similar values (IC_50_ of 8.09 ± 1.47 and 8.46 ± 1.86, respectively), while the effect of the **4b** on mitochondria was significantly higher than **4c** (Figure 7a,b). For this reason, one may infer that these compounds should be affecting something else in the parasites besides mitochondria that could explain this apparent discrepancy.

Considering this issue, we investigate another possible action of these compounds in these parasites. The acidocalcisomes are another relevant organelle characteristic of these trypanosomatids directly involved in bioenergetics and intracellular Ca^2+^ regulation. These are acidic vacuoles rich in calcium and phosphates, engaged in the production and storage of pyrophosphates, and alternative energetic currency in addition to ATP, well known in trypanosomatids [64]. For this reason, we also tested the possible effect of the compounds **4b**, **4c**, and **4e** on acidocalcisomes, using acridine orange accumulation as a well-recognized method [65]. This compound is released to the medium after parasite alkalization, denoting the loss of this organelle function. However, the results obtained indicated that none of these compounds showed any effect on the functionality of these organelles (results not shown), suggesting thus far that the impact on mitochondria is most relevant and confirms the mechanism of action of the compounds on these parasites. In this context, it is interesting to mention that in trypanosomatids, the disruption of intracellular Ca^2+^ homeostasis drives parasite death, thus proposing that alteration of this function could be a rational therapeutic target [64].

### 2.5. Cytotoxic Activity

We tested the cytotoxic effects of these compounds on eight cancer cell lines and two non-malignant cell lines. The IC_50_ values of the tested compounds were greater than 50 µM and were therefore considered inactive.

### 2.6. Statistical Analysis

This study used GraphPad Prism version 5.3 (GraphPad Prism Software Inc., La Jolla, CA, USA, 1992–2004) for statistical analysis. *p* values ≤ 0.05 were considered significant [66]. 

## 3. Materials and Methods

### 3.1. Chemistry

The products were examined using ^1^H and ^13^C NMR to verify their chemical structure. The spectra were recorded on a JEOL Eclipse^TM^ 270 (270 MHz/67.9 MHz) (JEOL Ltd., Tokyo, Japan) spectrometer using CDCl_3_ or DMSO-*d*_6_ as the solvents, and the spectra were reported in ppm downfield from the residual CHCl_3_ or DMSO (δ 7.25 or 2.50 for ^1^H NMR and 77.0 or 39.8 for ^13^C NMR, respectively). Thomas Hoover^TM^ (Thomas Scientific, Seattle, WA., USA) apparatus was used to measure melting points in open capillaries and are uncorrected. IR spectra were determined by Perkin-Elmer^TM^ Spectrum two FT-IR spectrometer (PerkinElmer Ltd., Buckinghamshire, UK) and are expressed in cm^−1^. An aluminium sheet covered with silica gel 60 F254 Merck^TM^ (Merck KGaA, Darmstadt, Germany) was used to monitor the reactions by thin layer chromatography (TLC). Compounds were visualized under UV light (254 nm). Column chromatography was performed on Merck silica gel 60 (40–63 µm) as a stationary phase. A Perkin ElmerTM 2400 (Perkin Elmer, Inc., Waltham, MA, USA) CHN elemental analyser was used to obtain the elemental analyses, and the results were within 0.4% of the prediction. Various chemical suppliers provided the chemicals and solvents; calcium hydride was used to dry dichloromethane (DCM).

### 3.2. Synthesis of 2-(7-Chloroquinolin-4-ylamino)ethanol (***2***)

Under an inert nitrogen atmosphere, 4,7-Dichloroquinoline **1** (5 g, 25 mmol) was combined with 2-Aminoethanol (24.3 g, 397 mmol) and a Triethylamine catalyst. The reaction mixture was refluxed for 12 h and then processed according to the published procedure [67,68]. Recrystallization from ethanol gave pure **2** as a white solid (95%); mp 213–215 °C, (Lit. 214 °C) [68,69]. IR (KBr) cm^−1^: 3350(-OH), 2950(C-H), 1588(-NH). ^1^H NMR (270 MHz, DMSO-*d*6) δ ppm: 3.32–3.39 (m, 2H, H_9_), 3.63–3.70 (m, 2H, H_10_), 4.88 (t, 1H, OH, *J* = 5.7 Hz), 6.50 (d, 1H, H_3_, *J* = 5.4 Hz), 7.28 (t, 1H, NH, *J* = 5.2 Hz), 7.45 (dd, 1H, H_6_, *J* = 2.2, 8.9 Hz), 7.78 (d, 1H, H_8_, *J* = 2.2 Hz), 8.26 (d, 1H, H_5_, *J* = 8.9 Hz), 8.39 (d, 1H, H_2_, *J* = 5.4 Hz). ^13^C NMR (68 MHz DMSO-*d*6) ppm: 45.7 (C_9_), 59.4 (C_10_), 99.3 (C_3_), 118.1, 124.6 (C_5_), 124.6 (C_6_), 128.1 (C_8_), 133.9, 149.7, 150.9, 152.4 (C_2_).

### 3.3. General Procedure for the Preparation of 2-(7-Chloroquinolin-4-ylamino)ethyl Benzoate Derivatives (***4a**–**m***)

In 10 mL of anhydrous DCM at 0 °C, benzoic acid corresponding **3a**–**m** (1 mmol), *N*-(3-Dimethylaminopropyl)-*N*’-ethylcarbodiimide hydrochloride (EDCI) (1 mmol) and 4-(Dimethylamino)-pyridine (DMAP) (0.4 mmol) were mixed under inert nitrogen atmosphere. After stirring for 30 min, the compound **2** was added (0.220 g, 1 mmol). Afterwards, the temperature was increased to room temperature (rt) and stirred for another 12 h. After washing the reaction mixture three times with NaHCO_3_, drying with Na_2_SO_4_, filtering, and removing it under reduced pressure, a solid was obtained, which was recrystallized from ethanol or chromatographed on silica gel with DCM and ethyl acetate as the eluent to yield the title compounds.

#### 3.3.1. 2-(7-Chloroquinolin-4-ylamino)ethyl-2-methoxybenzoate (**4a**)

Recrystallization from ethanol as a crystalline solid, yield 85%; m.p. 75–76 °C. IR (KBr) cm^−1^: 3225 (NH), 3024(C-H), 1700(C=O), 1520(-NH), 1265(C-O). ^1^H NMR (CDCl_3_, 270 MHz) *δ* ppm: 3.65 (dd, 2H, H9, *J* = 4.9, 10.4 Hz), 3.82 (s, 3H, OMe), 4.68 (t, 2H, H10, *J* = 5.4 Hz), 5.82 (br s, 1H, NH), 6.43 (d, 1H, H3, *J* = 5.4 Hz), 6.94–7.00 (m, 2H, H3′, 5′), 7.34 (dd, 1H, H6, *J* = 1.9, 8.9 Hz), 7.45–7.49 (m, 1H, H4′), 7.73 (d, 1H, H5, *J* = 8.9 Hz), 7.78 (dd, 1H, H6′, *J* = 1.7, 7.9 Hz), 7.94 (d, 1H, H8, *J* = 1.9 Hz), 8.51 (d, 1H, H2, *J* = 5.4 Hz). ^13^C NMR (CDCl_3_, 67.9 MHz) *δ* ppm: 43.1 (C9), 56.2 (OMe), 62.8 (C10), 99.2 (C3), 112.4 (C3’ or 5’), 117.4, 119.6, 120.4 (C3’ or 5’), 121.2 (C5), 125.6 (C6), 128.9 (C8), 131.8 (C6’), 134.1 (C4’), 135.1, 149.3, 149.7, 152.1 (C2), 159.3, 167.2 (C11). Anal. Calcd for C_19_H_17_ClN_2_O_3_: C, 63.96; H, 4.80; N, 7.85; O, 13.45. Found: C, 63.95; H, 4.80; N, 8.01; O, 13.49.

#### 3.3.2. 2-(7-Chloroquinolin-4-ylamino)ethyl-4-methoxybenzoate (**4b**)

Recrystallization from ethanol as a crystalline solid, yield 92%; m.p. 152–154 °C. IR (KBr) cm^−1^: 3232 (NH), 3022(C-H), 1705(C=O), 1577(-NH), 1372(-C-N), 1267(C-O). ^1^H NMR (CDCl_3_, 270 MHz) *δ* ppm: 3.68 (dd, 2H, H9, *J* = 5.2, 9.9 Hz), 3.84 (s, 3H, OMe), 4.69 (t, 2H, H10, *J* = 5.2 Hz), 6.06 (br s, 1H, NH), 6.43 (d, 1H, H3, *J* = 5.4 Hz), 6.90 (d, 2H, H3′, 5′, *J* = 8.9 Hz), 7.36 (dd, 1H, H6, *J* = 2.2, 8.9 Hz), 7.74 (d, 1H, H5, *J* = 8.9 Hz), 7.94 (d, 1H, H8, *J* = 2.2 Hz), 7.99 (d, 2H, H2′, 6′, *J* = 8.9 Hz), 8.48 (d, 1H, H2, *J* = 5.4 Hz). ^13^C NMR (CDCl_3_, 67.9 MHz) *δ* ppm: 43.5 (C9), 55.5 (OMe), 62.8 (C10), 98.9 (C3), 113.9 (C3’, 5’), 117.0, 121.6 (C5), 121.8, 125.9 (C6), 127.8 (C8), 131.9 (C2’, 6’), 135.7, 147.8, 150.4, 150.8 (C2), 164.0, 167.4 (C11). Anal. Calcd for C_19_H_17_ClN_2_O_3_: C, 63.96; H, 4.80; N, 7.85; O, 13.45. Found: C, 63.98; H, 4.82; N, 7.94; O, 13.43.

#### 3.3.3. 2-(7-Chloroquinolin-4-ylamino)ethyl-2,3-dimethoxybenzoate (**4c**)

Recrystallization from ethanol as a crystalline solid, yield 82%; m.p. 144–145 °C. IR (KBr) cm^−1^: 3222 (NH), 3017(C-H), 1702(C=O), 1580(-NH), 1376(-C-N), 1260(C-O). ^1^H NMR (CDCl_3_, 270 MHz) *δ* ppm: 3.69 (dd, 2H, H9, *J* = 4.9, 9.9 Hz), 3.83 (s, 3H, OMe), 3.87 (s, 3H, OMe), 4.69 (t, 2H, H10, *J* = 4.9 Hz), 5.98 (s, 1H, NH), 6.44 (d, 1H, H3, *J* = 5.5 Hz), 7.07–7.09 (m, 1H, H4′,5′), 7.28–7.32 (m, 1H, H6′), 7.37 (dd, 1H, H6, *J* = 2.2, 8.9 Hz), 7.79 (d, 1H, H5, *J* = 8.9 Hz), 7.95 (d, 1H, H8, *J* = 2.2 Hz), 8.51 (d, 1H, H2, *J* = 5.4 Hz). ^13^C NMR (CDCl_3_, 67.9 MHz) *δ* ppm: 43.0 (C9), 56.2 (OMe), 61.5 (OMe), 63.0 (C10), 99.0 (C3), 116.6 (C4’ or 5’), 117.3, 121.5 (C5), 122.3 (C6’), 124.1 (C5’ or 4’), 125.5, 125.8 (C6), 128.4 (C8), 135.4, 148.7, 149.3, 150.0, 151.5 (C2), 153.7, 167.1 (C11). Anal. Calcd for C_20_H_19_ClN_2_O_4_: C, 62.10; H, 4.95; N, 7.24; O, 16.54. Found: C, 62.09; H, 4.97; N, 7.41; O, 16.59.

#### 3.3.4. 2-(7-Chloroquinolin-4-ylamino)ethyl-2,4-dimethoxybenzoate (**4d**)

Recrystallization from ethanol as a crystalline solid, yield 85%; m.p. 90–92 °C. IR (KBr) cm^−1^: 3230 (NH), 3027(C-H), 1702(C=O), 1578(-NH), 1376(-C-N), 1232(C-O). ^1^H NMR (CDCl_3_, 270 MHz) *δ* ppm: 3.64 (dd, 2H, H9, *J* = 5.4, 10.9 Hz), 3.74 (s, 3H, OMe), 3.81 (s, 3H, OMe), 4.40 (t, 2H, H10, *J* = 5.4 Hz), 6.55 (dd, 1H, H5’, *J* = 2.2, 8.7 Hz), 6.60–6.63 (m, 2H, H3, 3’), 7.46 (dd, 1H, H6, *J* = 1.7, 8.7 Hz), 7.72 (d, 1H, H6’, *J* = 8.6 Hz), 7.79 (d, 1H, H8, *J* = 1.7 Hz), 8.26 (d, 1H, H5, *J* = 8.7 Hz), 8.41 (d, 1H, H2, *J* = 4.2 Hz). ^13^C NMR (CDCl_3_, 67.9 MHz) *δ* ppm: 40.5 (C9), 56.1 (OMe), 56.4 (OMe), 62.4 (C10), 99.5 (C3 or 3’), 99.6 (C3 or 3’), 105.9 (C5’), 112.4, 118.0, 124.5 (C5), 124.8 (C6), 128.2 (C8), 133.7 (C6’), 134.0, 149.7, 150.6, 152.5 (C2), 161.4, 164.6, 165.3 (C11). Anal. Calcd for C_20_H_19_ClN_2_O_4_: C, 62.10; H, 4.95; N, 7.24; O, 16.54. Found: C, 62.13, H, 5.01; N, 7.37; O, 16.61.

#### 3.3.5. 2-(7-Chloroquinolin-4-ylamino)ethyl-2,5-dimethoxybenzoate (**4e**)

Recrystallization from ethanol as a crystalline solid, yield 86%; m.p. 135–137 °C. IR (KBr) cm^−1^: 3225 (NH), 3019(C-H), 1709(C=O), 1584(-NH), 1210(C-O). ^1^H NMR (CDCl_3_, 270 MHz) *δ* ppm: 3.67–3.73 (m, 2H, H9), 3.75 (s, 3H, OMe), 3.77 (s, 3H, OMe), 4.68 (t, 2H, H10, *J* = 5,4 Hz), 6.53 (d, 1H, H3, *J* = 5.7 Hz), 6.90 (d, 1H, H3′, *J* = 9.1 Hz), 7.03 (dd, 1H, H4′, *J* = 3.0, 9.1 Hz), 7.30 (d, 1H, H6′, *J* = 3.0 Hz), 7.36 (dd, 1H, H6, *J* = 1.7, 9.2 Hz), 7.95–8.03 (m, 2H, H5, 8), 8.41 (d, 1H, H2, *J* = 5.7 Hz). ^13^C NMR (CDCl_3_, 67.9 MHz) *δ* ppm: 43.0 (C9), 55.9 (OMe), 56.9 (OMe), 63.0 (C10), 99.1 (C3), 114.3 (C3’), 116.3 (C6’), 117.3, 117.3, 119.3, 120.0, 120.1 (C4’), 121.3 (C5), 124.0, 125.7 (C6), 128.5 (C8), 149.9, 153.4, 153.6 (C2), 167.0 (C11). Anal. Calcd for C_20_H_19_ClN_2_O_4_: C, 62.10; H, 4.95; N, 7.24; O, 16.54. Found: C, 62.11; H, 4.94; N, 7.35; O, 16.59.

#### 3.3.6. 2-(7-Chloroquinolin-4-ylamino)ethyl-2,4,5-trimethoxybenzoate (**4f**)

Recrystallization from ethanol as a crystalline solid, yield 94%; m.p. 158–159 °C. IR (KBr) cm^−1^: 3237 (NH), 3023(C-H), 1712(C=O), 1580(-NH), 1353(-C-N), 1216(C-O). ^1^H NMR (CDCl_3_, 270 MHz) *δ* ppm: 3.72 (dd, 2H, H9, *J* = 5.2, 10.1 Hz), 3.83 (s, 3H, OMe), 3.84 (s, 3H, OMe), 3.93 (s, 3H, OMe), 4.68 (t, 2H, H10, *J* = 5.2 Hz), 6.48 (d, 1H, H3, *J* = 5.7 Hz), 6.51 (s, 1H, H3′), 7.34–7.38 (m, 2H, H6, 6′), 7.89 (d, 1H, H5, *J* = 8.9 Hz), 7.99 (d, 1H, H8, *J* = 1.9 Hz), 8.47 (d, 1H, H2, *J* = 5.7 Hz). ^13^C NMR (CDCl_3_, 67.9 MHz) *δ* ppm: 43.2(C9), 56.2(OMe), 56.7 (OMe), 57.2 (OMe), 62.5 (C10), 98.1 (C3’), 99.0 (C3), 110.1, 114.9 (C6’ or 6), 117.1, 121.8 (C5), 125.9 (C6’ or 6), 127.4 (C8), 135.9, 143.0, 150.5 (C2), 154.3, 156.1, 166.5 (C11). Anal. Calcd for C_21_H_21_ClN_2_O_5_: C, 60.51; H, 5.08; N, 6.72; O, 19.19. Found: C, 60.51; H, 5.11; N, 6.89; O, 19.23.

#### 3.3.7. 2-(7-Chloroquinolin-4-ylamino)ethyl-3,4,5-trimethoxybenzoate (**4g**)

Recrystallization from ethanol as a crystalline solid, yield 89%; m.p. 209–211 °C. IR (KBr) cm^−1^: 3224 (NH), 3030(C-H), 1692(C=O), 1510(-NH), 1376(-C-N), 1203(C-O). ^1^H NMR (CDCl_3_, 270 MHz) *δ* ppm: 3.71 (dd, 2H, H9, *J* = 4.9, 9.4 Hz), 3.86 (s, 6H, OMe), 3.88 (s, 3H, OMe), 4.70 (t, 2H, H10, *J* = 5.2 Hz), 6.06 (br s, 1H, NH), 6.45 (d, 1H, H3, *J* = 5.4 Hz), 7.26 (s, 2H, H2′, 6′), 7.35 (dd, 1H, H6, *J* = 2.2, 9.2 Hz), 7.76 (d, 1H, H5, *J* = 9.2 Hz), 7.94 (d, 1H, H8, *J* = 2.2 Hz), 8.49 (d, 1H, H2, *J* = 5.4 Hz). ^13^C NMR (CDCl_3_, 67.9 MHz) *δ* ppm: 43.3 (C9), 56.4 (OMe x 2), 61.0 (OMe), 63.2 (C10), 98.9 (C3), 107.5 (C2’, 6’), 117.2, 121.4 (C5), 124.3, 125.8 (C6), 128.3 (C8), 135.4, 143.2, 148.6, 150.1, 151.4 (C2), 153.2, 167.2 (C11). Anal. Calcd for C_21_H_21_ClN_2_O_5_: C, 60.51; H, 5.08; N, 6.72; O, 19.19. Found: C, 60.49; H, 5.13; N, 6.83; O, 19.21.

#### 3.3.8. 2-(7-Chloroquinolin-4-ylamino)ethyl-3-chlorobenzoate (**4h**)

Recrystallization from ethanol as a crystalline solid, yield 79%; m.p. 100–102 °C. IR (KBr) cm^−1^: 3239 (NH), 3020(C-H), 1700(C=O), 1579(-NH), 1255(C-O). ^1^H NMR (CDCl_3_, 270 MHz) *δ* ppm: 3.72 (dd, 2H, H9, *J* = 4.9, 9.9 Hz), 4.71 (t, 2H, H10, *J* = 5.1 Hz), 5.87 (br s, 1H, NH), 6.45 (d, 1H, H3, *J* = 5.2 Hz), 7.34–7.40 (m, 2H, H6, 5′), 7.52–7.56 (m, 1H, H4′), 7.73 (d, 1H, H5, *J* = 8.9 Hz), 7.89–7.93 (m, 1H, H6′), 7.94 (d, 1H, H8, *J* = 1.9 Hz), 7.99–8.01 (m, 1H, H2′), 8.52 (d, 1H, H2, *J* = 5.2 Hz). ^13^C NMR (CDCl_3_, 67.9 MHz) *δ* ppm: 43.2 (C9), 63.5 (C10), 99.0 (C3), 117.2, 121.2 (C5), 125.9 (C6), 127.9, 128.5 (C8), 128.9, 129.9, 133.6, 134.9, 135.4, 148.7, 149.8, 151.5 (C2), 151.6, 166.3 (C11). Anal. Calcd for C_18_H_14_Cl_2_N_2_O_2_: C, 59.85; H, 3.91; N, 7.76; O, 8.86. Found: C, 59.86; H, 3.94; N, 7.92; O, 8.93.

#### 3.3.9. 2-(7-Chloroquinolin-4-ylamino)ethyl-2-fluorobenzoate (**4i**)

Recrystallization from ethanol as a white solid, yield 80%; m.p. 110–111 °C. IR (KBr) cm^−1^: 3229 (NH), 2960(C-H), 1709(C=O), 1576(-NH), 1227(C-O). ^1^H NMR (CDCl_3_, 270 MHz) *δ* ppm: 3.84–3.88 (m, 2H, H9), 4.26 (br s, 1H, NH), 4.71 (t, 2H, H10, *J* = 5.4 Hz), 6.59 (d, 1H, H3, *J* = 4.7 Hz), 7.09–7.14 (m, 1H, H3′), 7.19 (m, 1H, H5′), 7.36 (dd, 1H, H6, *J* = 1.7, 8.9 Hz), 7.46–7.55 (m, 1H, H4′), 7.89 (m, 1H, H6′), 8.01 (s, 1H, H8), 8.27 (d, 1H, H5, *J* = 8.9 Hz), 8.34 (d, 1H, H2, *J* = 4.7 Hz). ^13^C NMR (CDCl_3_, 67.9 MHz) *δ* ppm: 42.7 (C9), 63.1 (C10), 98.5 (C3), 99.9, 117.0 (C3′), 117.3, 118.1, 121.6 (C5), 124.2 (C6), 126.8 (C8), 132.3, 135.1 (d, C2′, *J* = 48.1 Hz), 137.7, 146.6, 153.2 (C2), 160.1, 163.8, 164.8 (C11). Anal. Calcd for C_18_H_14_ClFN_2_O_2_: C, 62.71; H, 4.09; N, 8.13; O, 9.28. Found: C, 62.70, H, 4.11; N, 8.29; O, 9.35.

#### 3.3.10. 2-(7-Chloroquinolin-4-ylamino)ethyl-3,5-dimethylbenzoate (**4j**)

Column chromathography DCM:EtAc (4:1). White solid; yield: 97%; m.p. 178–179 °C. IR (KBr) cm^−1^: 3239 (NH), 2923(C-H), 2359(N-H), 1709(C=O), 1576(-NH), 1212(C-O). ^1^H NMR (CDCl_3_, 270 MHz) *δ* ppm: 2.34 (s, 6H, 2CH_3_), 3.69 (dd, 2H, H9, *J* = 4.9, 10.2 Hz), 4.69 (t, 2H, H10, *J* = 4.9 Hz), 5.87 (br s, 1H, NH), 6.45 (d, 1H, H3, *J* = 5.3 Hz), 7.20 (s, 1H, H4′), 7.36 (dd, 1H, H6, *J* = 2.0, 8.9 Hz), 7.65 (s, 2H, H2′, 6′), 7.72 (d, 1H, H5, *J* = 8.9 Hz), 7.95 (d, 1H, H8, *J* = 2.0 Hz), 8.55 (d, 1H, H2, *J* = 5.3 Hz). ^13^C NMR (CDCl_3_, 67.9 MHz) *δ* ppm: 21.3 (2 × CH_3_), 43.4 (C9), 63.1 (C10), 99.1 (C3), 117.4, 121.3 (C5), 125.7 (C6), 127.6 (C2’, 6’), 128.9 (C8), 129.4, 135.3, 138.4, 149.2, 149.7, 152.1 (C2), 168.1 (C11). Anal. Calcd for C_20_H_19_ClN_2_O_2_: C, 67.70; H, 5.40; N, 7.89; O, 9.02. Found: C, 67.71; H, 5.40; N, 8.11; O, 9.12.

#### 3.3.11. 2-(7-Chloroquinolin-4-ylamino)ethyl-5-methyl-2-nitrobenzoate (**4k**)

Column chromatography DCM:EtAc (3:2). Yellow solid; yield: 80%; m.p. 130–132 °C. IR (KBr) cm^−1^: 3223 (NH), 3015(C-H), 1733(C=O), 1521(C=N), 1512(NO_2_), 1340(CH_3_). ^1^H NMR (CDCl_3_, 270 MHz) *δ* ppm: 2.47 (s, 3H, CH), 3.72 (dd, 2H, H9, *J* = 5.3, 10.1 Hz), 4.66 (t, 2H, H10, *J* = 4.9 Hz), 5.56 (t, 1H, NH, *J* = 4.9 Hz), 6.42 (d, 1H, H3, *J* = 5.4 Hz), 7.40–7.45 (m, 2H, H6, 4′), 7.50 (d, 1H, H6′, *J* = 0.9 Hz), 7.84 (d, 1H, H5, *J* = 8.9 Hz), 7.86 (d, 1H, H3′, *J* = 8.3 Hz), 7.96 (d, 1H, H8, *J* = 2.1 Hz), 8.55 (d, 1H, H2, *J* = 5.4 Hz). ^13^C NMR (CDCl_3_, 67.9 MHz) *δ* ppm: 21.5 (C12), 42.2 (C9), 64.3 (C10), 99.0 (C3), 117.5, 121.6 (C5), 124.3 (C3’), 125.8 (C6), 127.6, 128.9 (C8), 130.6 (C6’), 132.5 (C4’), 135.2, 145.2, 149.4, 149.5, 152.1 (C2), 166.3 (C11). Anal. Calcd for C_19_H_16_ClN_3_O_4_: C, 59.15; H, 4.18; N, 10.89; O, 16.59. Found: C, 59.12; H, 4.19; N, 11.07; O 16.64.

#### 3.3.12. 2-(7-Chloroquinolin-4-ylamino)ethyl-4-tert-butylbenzoate (**4l**)

Column chromatography DCM:EtAc (9:1). White solid; yield: 60%; m.p. 182–183 °C. IR (KBr) cm^−1^: 3225 (NH), 3060(C-H), 2961(C-H), 2361(N-H), 1707(C=O), 1582(-NH), 1234(C-O). ^1^H NMR (CDCl_3_, 270 MHz) *δ* ppm: 1.31 (s, 9H, CH_3_), 3.66 (dd, 2H, H9, *J* = 4.9, 10.4 Hz), 4.70 (t, 2H, H10, *J* = 4.9 Hz), 5.81 (br s, 1H, NH), 6.43 (d, 1H, H3, *J* = 5.4 Hz), 7.36 (dd, 1H, H6, *J* = 1.9, 8.9 Hz), 7.45 (d, 2H, H3′, 5′, *J* = 8.4 Hz), 7.70 (d, 1H, H5, *J* = 8.9 Hz), 7.94 (d, 1H, H8, *J* = 1.9 Hz), 7.97 (d, 2H, H2′, 6′, *J* = 8.4 Hz), 8.53 (d, 1H, H2, *J* = 5.4 Hz). ^13^C NMR (CDCl_3_, 67.9 MHz) *δ* ppm: 31.1 (CH_3_), 35.2, 43.4 (C9), 62.9 (C10), 99.0 (C3), 117.2, 121.2 (C5), 125.6 (C3’, 5’), 125.7 (C6), 126.6, 128.8 (C8), 129.7 (C2’, 6’), 135.1, 149.1, 149.6, 152.0 (C2), 157.5, 167.6 (C11). Anal. Calcd for C_22_H_23_ClN_2_O_2_: C, 69.01; H, 6.05; N, 7.32; O, 8.36. Found: C, 69.03; H, 6.07; N, 7.49; O, 8.38.

#### 3.3.13. 2-(7-Chloroquinolin-4-ylamino)ethyl-4-(trifluoromethyl)benzoate (**4m**)

Column chromatography DCM:EtAc (4:1). Red solid; yield: 81%; m.p. 164–166 °C. IR (KBr) cm^−1^: 3231 (NH), 3054(C-H), 2929(C-H), 2358(N-H), 1729(C=O), 1584(-NH), 1326(C-F_3_), 1241(C-O). ^1^H NMR (CDCl_3_, 270 MHz) *δ* ppm: 3.73 (dd, 2H, H9, *J* = 5.1, 10.4 Hz), 4.74 (t, 2H, H10, *J* = 5.1 Hz), 5.85 (br s, 1H, NH), 6.45 (d, 1H, H3, *J* = 5.4 Hz), 7.36 (dd, 1H, H6, *J* = 2.1, 8.9 Hz), 7.70 (m, 3H, H5, 3′, 5′), 7.94 (d, 1H, H8, *J* = 2.1 Hz), 8.15 (d, 2H, H2′, 6′, *J* = 8.1 Hz), 8.53 (d, 1H, H2, *J* = 5.3 Hz). ^13^C NMR (CDCl_3_, 67.9 MHz) *δ* ppm: 43.1 (C9), 63.6 (C10), 99.0 (C3), 117.2, 121.3 (C5), 125.6, 125.7 (c, CF_3_, *J* = 15.0 Hz), 128.7 (C8), 130.3 (C2′, 6′),132.7 (d, *J* = 4.7 Hz), 134.9, 135.3 (c, *J* = 127.6 Hz), 148.9, 149.7, 151.8 (C2), 166.4 (C11). ^19^F NMR (CDCl_3_) *δ* ppm: −63.18 ppm. Anal. Calcd for C_19_H_14_ClF_3_N_2_O_2_: C, 57.81; H, 3.57; N, 14.44; O, 8.11. Found: C 57.78; H, 3.56; N, 14.69; O, 8.17.

### 3.4. ADME/Tox Profile Prediction

Based on Lipinski and Veber rules, the drug-likeness properties of the compounds **2**, **4a**–**m** were determined through the *SwissADME* program, a free online program available at http://www.swissadme.ch/ (accessed on 15 March 2023) [47,48]. The site (http://structure.bioc.cam.ac.uk/pkcsm) also offers free access to *pkCSM-pharmacokinetics*, a method for predicting and optimizing small molecules’ ADME/Tox properties [51].

### 3.5. Inhibition of β-Hematin Formation

The β-hematin (β-H) formation assay was performed as described previously [59]. In 96-well microplates, hemin chloride solution (50 μL, 4 mM) was dissolved in dimethyl sulfoxide (DMSO) (5 mg mL^−1^). We dissolved the compounds in DMSO and added different concentrations (100–5 mM) to the test wells (50 μL). As controls, 50 μL of water and 50 μL of DMSO were used. Experiments were conducted in triplicate. In order to generate β-H, acetate buffer (100 μL, 0.2 M, pH 4.4) was used. After 48 h at 37 °C, the plates were centrifuged (4000 RPM for 15 min, IEC-CENTRA, MP4R) (International Equipment Company, MA, USA). After discarding the supernatants, the pellet was washed twice with DMSO (200 μL) and dissolved in NaOH (200 μL, 0.2 N). Solubilizing the aggregates with NaOH (0.1 N) further allowed us to measure their absorbance at 405 nm (BIORAD-550 microplate reader) (BIO-RAD Laboratories, Hercules, CA, USA). The outcomes are expressed as percentages of inhibition of β-H formation [60].

### 3.6. Parasite, Experimental Host and Strain Maintenance

For the experiments, male BALB/c mice, 18–22 g, were fed a commercial pellet diet under controlled conditions approved by the Institute of Immunology’s Ethics Committee. The animals were infected with the rodent malaria ANKA strain of *Plasmodium berghei*; 1 × 10^6^ infected erythrocytes diluted in phosphate-buffered saline (PBS; 10 mM, pH 7.4, 0.1 mL) were injected intraperitoneally into mice. Microscopic examination of Giemsa-stained smears was used to monitor parasitaemia [20,60].

### 3.7. Four-Day Suppressive Test

*Plasmodium berghei*-infected red blood cells were injected into the caudal vein of BALB/c mice (18–23 g) (n = 6). Within two hours of infection, the active compounds from the in vitro test began to be administered (inhibition of β-H formation). Following dissolution in DMSO (0.1 M), the compounds were diluted with 2% saline-Tween 20 solution. The compounds (dose 25 mg kg^−1^) were administered ip for four days. Giemsa-stained smears were examined on the fourth day to determine the parasite load. As a positive control, CQ (25 mg kg^−1^) in DMSO (0.1 M) diluted with 2% saline-Tween 20 solution was used. A control group of mice infected with *P. berghei* incubated with saline solution was used to determine survival times. The results are expressed as a percentage of parasitaemia, and the survival curve was constructed based on the number of days of survival after treatment with the compound over that of mice infected without treatment [20,60,61].

### 3.8. In Vitro Toxicity on Mouse Red Blood Cells (RBCs)

To evaluate the in vitro toxicological effects of the compounds, we measured the haemoglobin released into the supernatant fraction of lysed red blood cells (RBCs) [62]. A spectrophotometer at 550 nm was used to measure the haemoglobin released. To obtain 100% RBCs, mouse blood was spun down at 800 g for 10 min and washed thrice with saline solution. A 2% final suspension of RBCs was incubated with the synthesized compounds (1 mM) at 37 °C for 45 min. Chloroquine was used as the standard drug in the same concentration as the test compounds. RBCs were lysed in 1% saponin solution and were used as a positive control and a PBS solution as a negative control. The percentage of haemolysis was calculated using the formula: % Haemolysis = [(AT − AN)/(AP − AN)] × 100, where AT = mean absorbance of the test compound; AN = mean absorbance of the negative control; AP = mean absorbance of the positive control.

### 3.9. Cell Culture

The cancer cell line was purchased from the American Type Tissue Culture Collection (ATCC) (Manassas, VA, USA) and maintained as previously described [69]. Cell lines: A549 (lung adenocarcinoma), U2OS (osteosarcoma), CCRF-CEM (childhood T cell acute lymphoblastic leukaemia), a daunorubicin-resistant subline of CCRF-CEM cells (CEM-DNR bulk), HCT116 (colorectal carcinoma), HCT116p53-/- (HCT116 with deleted p53 gene), K562 (chronic myeloid leukaemia), K562-TAX (chronic myeloid leukaemia paclitaxel-resistant subline), and the two non-malignant cell lines BJMRC-5 (human lung fibroblasts), and BJ (human fibroblasts from foreskin). The cells were cultivated in DMEM/RPMI 1640 medium of 10 g/L glucose, 100 U/mL, 100 mg/mL streptomycin, penicillin, 2 mM glutamine, 10% foetal calf serum, and NaHCO_3_ was used to maintain cell lines in Nunc/Corning 80 cm^2^ plastic flasks and culture them according to ATCC or Horizon recommendations.

### 3.10. Cytotoxic Activity

A robotic platform (High-ResBiosolutions) (Manchester, M17 1RW, UK) was used at the Institute of Molecular and Translational Medicine to determine cell viability for the compounds **2**, **4a**–**m**-treated cell lines. Suspensions of cells were made and diluted according to their type and added by automatic pipettor (30 µL) to microtiter plates with 384 wells. The compounds were dissolved in DMSO and then in media to a final concentration of 10 and 50 µM. The final concentration of sterile DMSO in cell culture was 0.1%, not affecting cell viability or metabolism [69]. At zero time, 0.15 µL of the cell mixture was added to the microtiter plate wells using an echo acoustic non-contact liquid handler Echo550 (Labcyte). The experiments were performed in triplicate and incubated at 37 °C for 72 h in a 5% CO_2_ atmosphere at 100% humidity. The 3-(4,5-dimethylthiazol-2yl)-5-(3-carboxymethoxyphenyl)-2H-tetrazolium (MTS) was used to assess the cells as described by the manufacturer. A portion (5 µL) of the MTS stock solution was pipetted into each well and incubated for an additional 1 to 4 h. A measurement was made of the optical density 490 (OD491). Cell viability was calculated as follows: Tumour cell survival (TCS) = (ODcompound-exposed well/mean OD control wells) × 100%. The control cells were treated with DMSO. Using Dotmatics software 2.0, the IC_50_ value, the concentration of the compound required to cause 50% of cell death, was calculated [69,70].

### 3.11. Culture of L. mexicana Promastigotes and Growth Inhibition Experiments

The growth of *L. mexicana* was carried out as previously reported [63]. *L. mexicana* (Bel 21 strain) promastigotes were cultured in liver infusion-tryptose (LIT) medium supplemented with 10% foetal bovine serum using continuous agitation at 29 °C, as reported previously [63,64]. A growth curve was performed to evaluate the susceptibility of parasites to the different compounds at different concentrations. Parasites were counted daily using a Neubauer chamber. The initial parasite concentration was 10^6^ parasites/mL, and either the drug or DMSO was added after 24 h. The final concentration of sterile DMSO in the parasite culture was 0.1%, not affecting the viability. At least three independent experiments were performed for each compound and dose, and the 50% inhibitory concentrations (IC_50_s) were determined. FCCP (2 µM) was used as a positive control.

### 3.12. Determination of Mitochondrial Membrane Potential of Leishmania mexicana

The effect of different compounds on the mitochondrial membrane potential of L. mexicana promastigotes was evaluated using rhodamine 123, a fluorescent dye distributed inside mitochondria according to the electrochemical membrane potential, as reported previously [64,65]. Briefly, 10^8^ parasites were obtained by centrifugation at 600× *g* for 2 min and washed in phosphate-buffered saline solution (PBS) plus 1% glucose. The pellet was resuspended in the same buffer but in the presence of 20 µM rhodamine 123 for 45 min at 29 °C in the dark with mild agitation. Then, parasites were washed twice, resuspended in the same buffer, and transferred into a stirred cuvette. Measurements (excitation wavelength [λex], 488 nm; emission wavelength [λem], 530 nm) were made in a Hitachi 7000 spectrofluorometer at 29 °C (Hitachi High-Tech Corporation, Tokyo, Japan). FCCP (2 µM) was used as a positive control.

## 4. Conclusions

In summary, a series of heterocyclic chloroquine hybrids were obtained, each from a two-step synthetic pathway, with overall yields of up to 60% in most cases. All compounds were characterized by IR, ^1^H NMR, ^13^C NMR, and by elemental analysis. In vitro, all compounds significantly reduced haeme crystallization with an IC_50_ < 10 µM; however, in vivo, the reduction in parasitaemia and survival time increase were marginal as antimalarials, with only two compounds, **4c** and **4e**, giving survival times of 16.71 ± 2.16 and 14.43 ± 1.20 days, respectively. The ADME/Tox analysis predicted moderate lipophilicity values, low unbound fraction values, and a poor distribution for these compounds; therefore, moderate bioavailability was expected. Regarding the possible mechanism of action of the three compounds **4b**, **4c**, and **4e** evaluated on *L. mexicana,* we conclude that the observed leishmanicidal activity is a product of the collapse of the parasite mitochondrial electrochemical membrane potential (Δφ). We also tested the possible effect of these three compounds on acidocalcisomes. The results indicated that none of these compounds showed any effect on the functionality of these organelles, suggesting that the effect on mitochondria is more relevant thus far, turning it into an interesting rational therapeutic objective. This class of hybrids also showed a low cytotoxicity against mammalian cancerous and noncancerous human cell lines. Concerning metabolism, except for **2**, these compounds may be inhibitors for the evaluated enzyme isoforms (CYPs), which are a determinant in biotransformation processes. To complement our in silico evaluation, we calculated other molecular descriptors, such as the topological polar surface area, according to Veber’s rules, except **2** and **4i**; the rest of the compounds violated this descriptor. Finally, these descriptors allowed us to understand our compounds’ marginal or low activity in vivo.

## Data Availability

Data are contained within the article and the Appendix A.

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
