# Peer review of "Synthesis, Antimalarial, Antileishmanial, and Cytotoxicity Activities and Preliminary In Silico ADMET Studies of 2-(7-Chloroquinolin-4-ylamino)ethyl Benzoate Derivatives"

_pharmaceuticals, 2023, doi:10.3390/ph16121709_

Round 1
Reviewer 1 Report
Comments and Suggestions for Authors
Gutierrez et al., synthesized new derivatives of original malaria drug,
quinoline and got 2 potential malaria drugs. It is important to have alternative drugs, which this study attempt to do.
Why they are doing this need to be justified: if their performance is not that much better than the mother drug, what is the added value, is that cost, or could be we need alternative drugs in case of drug resistance need to be justified and discussed. Improve the introduction by articulating the story clearly. See specific comments in the document directly as attached.
When we say low toxicity how low
positive control for leishmania
antimalarial activity. For each comparison add statistics
table 4 only present the one that has data.

Comments on the Quality of English LanguageAuthor Response
Comments and Suggestions for Authors
Gutierrez et al., synthesized new derivatives of original malaria drug,
quinoline and got 2 potential malaria drugs. It is important to have alternative drugs, which this study attempt to do.
Why they are doing this need to be justified: if their performance is not that much better than the mother drug, what is the added value, is that cost, or could be we need alternative drugs in case of drug resistance need to be justified and discussed. Improve the introduction by articulating the story clearly. See specific comments in the document directly as attached.
The observations described in the manuscript have been accepted and the modifications have been made, in the manuscript it is marked in yellow.
Pag. 2, lines 66 and 69
Pags. 2 and 3, lines 85-97
Pag.3, lines 101, 103
Pag. 3, line 109
Pag. 4, lines 147-149
Pag. 4, lines 159-160
Pag. 5, lines 182-183
Pag. 5, lines 186-189
Pag. 6, line 193
Pag. 6, Table 1 compound 2
Pag. 6, lines 201, 204-207
Pag. 7, line 214 figure resolution
Pag. 8, line 222
Pag. 8, lines 232, 246-247
Pag. 9, lines 251-252, 256-258, 266-268, 271-273 and 282-284
Pag. 10, Lines 296,297, 306
Pag. 11, lines 331-336, 343, 345, 351
Pag. 12, lines 360, 377, 378,
Pag. 13, lines 403-414
Pag. 18, lines 621-627, 629-636, 646-648, 651-653
Pag. 19, lines 656, 666-667, 681
Pag. 20, line 714
- When we say low toxicity how low
- When the IC50 values in these assays are above 50 uM.
- positive control for leishmania
-Pag 11, line 333-336, and in table 5.
- antimalarial activity. For each comparison add statistics
-Chloroquine is the positive control and the statistics are incorporated in table 4 and pag. 10 lines 296-297, Pag. 11, lines 331-332.
- table 4 only present the one that has data.
- Page 10, Table 4 was modified with the suggested observation.
The doubt is valid, however we do not use deuterated solvents with TMS traces, we work and the programs that come with the equipment are programmed to use the traces of non-deuterated solvent as a reference. I hope this clarifies your comment.
Reviewer 2 Report
Comments and Suggestions for Authors
The manuscript entitled: " Synthesis, antimalarial, antileishmanial, cytotoxicity activities, and preliminary in silico ADMET studies of 2-(7-chloroquinolin-4-ylamino)ethyl benzoate derivatives" comprises the necessary elements of scientific novelty. The results are interesting, clearly presented including discussion and supported by the literature properly. I recommend that the manuscript should be published in its current form.
Author Response
Thank you for the recommendation.
Reviewer 3 Report
Comments and Suggestions for Authors In my opinion, although the authors have made a significant effort in presented work, the manuscript in this form should not be accepted for publication before it’s revision.It is necessary to revise:
3. Materials and methods
3.8. In vitro toxicity in mouse red blood cells (RBC)
The authors should clarify how the tested compounds were prepared for the mouse red blood cell (RBC) toxicity tests.
3.9. Cell culture
Authors should describe which cells were used in the experiments and how they were obtained. What equipment was used for the cell cultures.
3.10. Cytotoxic activity
It should be clarified whether all dilutions were prepared using DMSO as a solvent. From the sentence "The compounds (10 and 50 μM) were dissolved in 100% DMSO, at time zero 0.15 μL was added to the microtiter plate..." (line 597), it can be concluded that the tested compounds were dissolved in DMSO for all concentrations used. In this case, the question arises as to the objectivity of the results obtained with regard to the toxicity of the compounds tested, while it is generally known that only 0.1% DMSO is considered safe for almost all cells. 0.5% DMSO as a final concentration is generally used for cell cultures without cytotoxicity. 1% DMSO does not cause toxicity in some cells, but 0.5% DMSO is recommended.
The authors should explain why were the control cells treated with 100% DMSO?
In addition, cells growing normally in suspension and cells growing in monolayers were included in the experiments. All cells in the cytotoxicity test were used in suspension. It is necessary to explain why it was done that way and provide evidence that it has no effect on the results obtained.
In general, the names, manufacturers and manufacturing locations of the devices used are missing.
Results :
Considering the ambiguities related to the use of DMSO as a solvent for the tested compounds, it is necessary that the authors first clarify the methodology used in order to assess the objectivity of the results obtained.
References:References 5, 7, 26, and 71 are not written in accordance with the instruction (https://mdpi-res.com/data/mdpi_references_guide_v5.pdf ).
Comments on the Quality of English LanguageModerate editing of English language required.
Author Response
Comments and Suggestions for Authors
In my opinion, although the authors have made a significant effort in presented work, the manuscript in this form should not be accepted for publication before it’s revision.
It is necessary to revise:
- Materials and methods
3.8. In vitro toxicity in mouse red blood cells (RBC)
- The authors should clarify how the tested compounds were prepared for the mouse red blood cell (RBC) toxicity tests.
-Pag. 18, lines 621-627, in yellow the recommendation has been incorporated.
3.9. Cell culture
- Authors should describe which cells were used in the experiments and how they were obtained. What equipment was used for the cell cultures.
-Pag. 18, lines 629-636, highlighted in yellow, the recommendation has been incorporated
3.10. Cytotoxic activity
- It should be clarified whether all dilutions were prepared using DMSO as a solvent. From the sentence "The compounds (10 and 50 μM) were dissolved in 100% DMSO, at time zero 0.15 μL was added to the microtiter plate..." (line 597), it can be concluded that the tested compounds were dissolved in DMSO for all concentrations used. In this case, the question arises as to the objectivity of the results obtained with regard to the toxicity of the compounds tested, while it is generally known that only 0.1% DMSO is considered safe for almost all cells. 0.5% DMSO as a final concentration is generally used for cell cultures without cytotoxicity. 1% DMSO does not cause toxicity in some cells, but 0.5% DMSO is recommended.
The authors should explain why were the control cells treated with 100% DMSO?
In addition, cells growing normally in suspension and cells growing in monolayers were included in the experiments. All cells in the cytotoxicity test were used in suspension. It is necessary to explain why it was done that way and provide evidence that it has no effect on the results obtained.
-We would like to thank the reviewer for the useful comments. We apologize for the mistake concerning the unclear procedure of DMSO dilution for cytotoxic assays. The compounds were diluted in DMSO, and then diluted in media DMEM until the concentrations for the assay were reached. At 50 µM concentration, the amount of total DMSO is 0.1 %. Thus the assay of all the cell lines were performed at this concentration. The description of the cell lines was in the results section, we apologize for the mistake, and was transferred to Material and Methods. The compounds did not affect cell viability in normal cells MRC 5 and BJ and it also did not affect the different cancer cell lines, lung epithelial, osteosarcoma, colon cancer, T cell lymphoma and leukemia.
Pags. 18, 19, lines 646-648, 651-653 you can see that the modifications were incorporated.
- In general, the names, manufacturers and manufacturing locations of the devices used are missing.
-Thanks for the observation, the sites where the equipment has been manufactured have been incorporated.
Results :
Considering the ambiguities related to the use of DMSO as a solvent for the tested compounds, it is necessary that the authors first clarify the methodology used in order to assess the objectivity of the results obtained.
References:
- References 5, 7, 26, and 71 are not written in accordance with the instruction (https://mdpi-res.com/data/mdpi_references_guide_v5.pdf).
- References have been described following the guidelines of the journal.
Comments on the Quality of English Language
Round 2
Reviewer 3 Report
Comments and Suggestions for Authors
The manuscript has been considerably improved.
Typographical, syntactical, and grammatical errors still need to be corrected in the manuscript.
Comments on the Quality of English LanguageTypographical, syntactical, and grammatical errors still need to be corrected in the manuscript.